# PROGRAM SYNTHESIS WITH PRIORITY QUEUE TRAINING

**Daniel A. Abolafia, Mohammad Norouzi, Jonathan Shen, Rui Zhao, Quoc V. Le**
`{danabo, mnorouzi, jonathanasdf, rzhao, qvl}@google.com`
Google Brain

## ABSTRACT

We consider the task of program synthesis in the presence of a reward function over the output of programs, where the goal is to find programs with maximal rewards. We introduce an iterative optimization scheme, where we train an RNN on a dataset of $K$ best programs from a priority queue of the generated programs so far. Then, we synthesize new programs and add them to the priority queue by sampling from the RNN. We benchmark our algorithm, called priority queue training (or PQT), against genetic algorithm and reinforcement learning baselines on a simple but expressive Turing complete programming language called BF. Our experimental results show that our simple PQT algorithm significantly outperforms the baselines. By adding a program length penalty to the reward function, we are able to synthesize short, human readable programs.

## 1 INTRODUCTION

Automatic program synthesis is an important task with many potential applications. Traditional approaches (*e.g.,* Muggleton & de Raedt (1994); Angulin (1987)) typically do not make use of machine learning and therefore require domain specific knowledge about the programming languages and hand-crafted heuristics to speed up the underlying combinatorial search. To create more generic programming tools without much domain specific knowledge, there has been a surge of recent interest in developing neural models that facilitate some form of memory access and symbolic reasoning (*e.g.,* Reed & de Freitas (2016); Neelakantan et al. (2016); Kaiser & Sutskever (2016); Zaremba et al. (2016); Graves et al. (2016)). Despite several appealing contributions, none of these approaches is able to synthesize source code in an expressive programming language.

More recently, there have been several successful attempts at using neural networks to explicitly induce programs from input-output examples (Riedel et al., 2016; Bunel et al., 2016; Balog et al., 2017; Parisotto et al., 2016) and even from unstructured text (Parisotto et al., 2016), but often using restrictive programming syntax and requiring supervisory signal in the form of ground-truth programs or correct outputs. By contrast, we advocate the use of an expressive programming language called BF[1], which has a simple syntax, but is Turing complete. Moreover, we aim to synthesize programs under the reinforcement learning (RL) paradigm, where only a solution checker is required to compute a reward signal. Furthermore, one can include a notion of code length penalty or execution speed into the reward signal to search for short and efficient programs. Hence, the problem of program synthesis based on reward is more flexible than other formulations in which the desired programs or correct outputs are required during training.

To address program synthesis based on a reward signal, we study two different approaches. The first approach is a policy gradient (PG) algorithm (Williams, 1992), where we train a recurrent neural network (RNN) to generate programs one token at a time. Then, the program is executed and scored, and a reward feedback is sent back to the RNN to update its parameters such that over time better programs are produced. The second approach is a deceptively simple optimization algorithm called *priority queue training (PQT)*. We keep a priority queue of $K$ best programs seen during training and train an RNN with a log-likelihood objective on the top $K$ programs in the queue. We then sample new programs from the RNN, update the queue, and iterate. We also compare against a genetic

---

[1] https://en.wikipedia.org/wiki/Brainfuck

algorithm (GA) baseline which has been shown to generate BF programs Becker & Gottschlich (2017). Surprisingly, we find that the PQT approach significantly outperforms the GA and PG methods.

We assess the effectiveness of our method on the BF programming language. The BF language is Turing complete, while comprising only 8 operations. The minimalist syntax of the BF language makes it easier to generate a syntactically correct program, as opposed to more higher level languages. We consider various string manipulation, numerical, and algorithmic tasks. Our results demonstrate that all of the search algorithms we consider are capable of finding correct programs for most of the tasks, and that our method is the most reliable in that it finds solutions on most random seeds and most tasks.

The key contributions of the paper include,

- We propose a learning framework for program synthesis where only a reward function is required during training (the ground-truth programs or correct outputs are not needed). Further, we advocate to use a simple and expressive programming language, BF, as a benchmark environment for program synthesis (see also Becker & Gottschlich (2017)).
- We propose an effective search algorithm using a priority queue and an RNN.
- We propose an experimental methodology to compare program synthesis methods including genetic algorithm and policy gradient. Our methodology measures the success rates of each synthesis method on average and provides a standard way to tune the hyper-parameters. With this methodology, we find that a recurrent network trained with priority queue training outperforms the baselines.

## 2 RELATED WORK

Our method shares the same goal with traditional techniques in program synthesis and inductive programming (Summers, 1977; Biermann, 1978; Muggleton & de Raedt, 1994; Angulin, 1987). These techniques have found many important applications in practice, ranging from education to programming assistance (Gulwani, 2010). In machine learning, probabilistic program induction has been used successfully in many settings, such as learning to solve simple Q&A (Liang et al., 2010), and learning with very few examples (Lake et al., 2015).

There has been a surge of recent interest in using neural networks to induce and execute programs either implicitly or explicitly (Graves et al., 2014; Zaremba & Sutskever, 2014; Joulin & Mikolov, 2015; Kaiser & Sutskever, 2016; Kurach et al., 2015; Neelakantan et al., 2016; Reed & de Freitas, 2016; Andreas et al., 2016; Balog et al., 2017; Bhoopchand et al., 2016; Gaunt et al., 2016; Khanh Dam et al., 2016; Neelakantan et al., 2017; Riedel et al., 2016; Schaechtle et al., 2016; Zaremba et al., 2016; Miceli Barone & Sennrich, 2017; Beltramelli, 2017; Cai et al., 2017; Devlin et al., 2017; Hu et al., 2017; Guu et al., 2017; Johnson et al.; Liang et al., 2017; Murali et al., 2017; Parisotto et al., 2016; Rabinovich et al., 2017; Vigueras et al., 2017; Yin & Neubig, 2017). For example, there have been promising results on the task of binary search (Nachum et al., 2017a), sorting an array of numbers (Reed & de Freitas, 2016; Cai et al., 2017), solving simple Q&A from tables (Neelakantan et al., 2016; 2017), visual Q&A (Andreas et al., 2016; Hu et al., 2017; Johnson et al.), filling missing values in tables (Devlin et al., 2017), and querying tables (Liang et al., 2017). There are several key components that highlight our problem formulation in the context of previous work. First, our approach uses a Turing complete language instead of a potentially restricted domain-specific language. Second, it does not need existing programs or even the stack-trace of existing programs. Third, it only assumes the presence of a verifier that scores the outputs of hypothesis programs, but does not need access to correct outputs. This is important for domains where finding the correct outputs is hard but scoring the outputs is easy. Finally, our formulation does not need to modify the internal workings of the programming language to obtain a differentiable error signal.

The PG approach adopted in this paper for program synthesis is closely related to neural architecture search (Zoph & Le, 2017) and neural combinatorial optimization (Bello et al., 2016), where variants of PG are used to train an RNN and a pointer network (Vinyals et al., 2015) to perform combinatorial search. Nachum et al. (2017a) applies PG to program synthesis, but they differ from us in that they train RNNs that implicitly model a program by consuming inputs and emitting machine instructions as opposed to explicit programs. Our PG baseline resembles such previous techniques.

The PQT algorithm presented here is partly inspired by Liang et al. (2017), where they use a priority queue of top-$K$ programs to augment PG with off-policy training. PQT also bears resemblance to the cross-entropy method (CEM), a reinforcement learning technique which has been used to play games such as Tetris (Szita & Lörincz, 2006).

Our use of BF programming language enables a comparison between our technique and a concurrent work by Becker & Gottschlich (2017) on the use of genetic algorithms for program synthesis in the BF language. However, our method for program synthesis has important benefits over Becker & Gottschlich (2017) including better performance and the potential for transfer learning, which is possible with neural networks (Johnson et al., 2016). We also make the observation that PQT alone is stable and effective, without needing to use PG.

## 3 APPROACH

We implement a generative model of programs as an RNN that emits a strings of BF language one character at a time. Figure 2 depicts the RNN model, which enables sampling a sequence of BF characters in an autoregressive fashion, where one feeds the previous prediction as an input to the next time step. The input to the first time step is a special START symbol. The RNN stops when it generates a special EOS symbol, indicating the end of sequence, or if the length of the program exceeds a pre-specified maximum length. The predictions at each timestep are sampled from a multinomial distribution (a softmax layer with shared weights across timesteps). The joint probability of the program sequence is the product of the probabilities of all of the tokens.

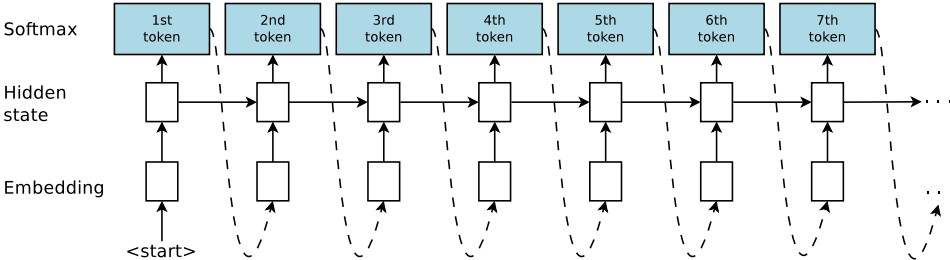

Figure 1: An overview of our synthesizer. The synthesizer is an RNN, which generates the program in an autoregressive fashion.

We study two training algorithms, which are also compatible and can be combined. These are policy gradient, and priority queue training. We treat the RNN program synthesizer as a policy $\pi(a_{1:T} \, ; \, \theta)$ parametrized by $\theta$, where $a_{1:T} \equiv (a_1, \ldots, a_T)$ denotes a sequence of $T$ actions, each of which represents a symbol in the BF langauge (and optionally an EOS symbol). The policy is factored using the chain rule as $\pi(a_{1:T} \, ; \, \theta) = \prod_t \pi(a_t \mid a_{1:t-1} \, ; \, \theta)$ where each term in the product is given by the RNN as depicted in Figure 2. Typically an RL agent receives a reward after each action. However in our setup, we cannot score the code until completion of the program (either by emitting the EOS symbol or by hitting the maximum episode length), and accordingly, only a terminal reward of $r(a_{1:T})$ is provided. The goal is to learn a policy that assigns a high probability to plausible programs.

### 3.1 POLICY GRADIENT (PG)

We use the well known policy gradient approach: the REINFORCE algorithm (Williams, 1992). As suggested by REINFORCE, we optimize the parameters $\theta$ to maximize the following expected reward objective,

$$O_{\text{ER}}(\theta) \; = \; \mathbb{E}_{\pi(a_{1:T} \, ; \, \theta)}[r(a_{1:T})] \, . \tag{1}$$

We perform stochastic gradient descent in $O_{\text{ER}}$ to iteratively refine $\theta$. To estimate the gradient of (1), we draw $N$ Monte Carlo samples from $\pi_\theta$ denoted $\{a^i_{1:T_i}\}^N_{i=1}$ and compute the gradient of the policy as,

$$\nabla_\theta O_{\text{ER}}(\theta) \; \approx \; \frac{1}{N} \sum_{i=1}^{N} \left[ \left(r(a^i_{1:T_i}) - b\right) \sum_{t=1}^{T_i} \nabla_\theta \log \pi(a^i_t \mid a^i_{1:t-1} \, ; \, \theta) \right] \, , \tag{2}$$

where $N$ is the number of episodes sampled from the policy in one mini-batch, and $T_i$ denotes the number of actions in the $i^{\text{th}}$ episode.

The gradient in Equation (2) is an unbiased estimate of the policy's true gradient, but it suffers from high variance in practice. The term $b$, known as a baseline, subtracted from the rewards serves as a control variate to reduce the variance of this estimator (Williams, 1992). We use an exponential moving average over rewards as the baseline.

## 3.2 PRIORITY QUEUE TRAINING (PQT)

Our key technical contribution in the paper involves training an RNN with a continually updated buffer of top-$K$ best programs (*i.e.,* a priority queue of maximum size $K$). The queue is initialized empty, and after each gradient update, it is provided with new sampled programs, keeping only the programs that fall within the $K$ highest rewarded programs. We use supervised learning to maximize the probability of the programs in the top-$K$ buffer denoted $\{\tilde{a}_{1:\tilde{T}_k}^k\}_{k=1}^K$ under the current policy. In this way, the RNN and priority queue bootstrap off each other, with the RNN finding better programs through exploration, and the priority queue providing better training targets. The objective for PQT is simply log-likelihood in the form,

$$O_{\text{TOPK}}(\theta) \;=\; \frac{1}{K} \sum_{k=1}^{K} \log \pi(\tilde{a}_{1:\tilde{T}_k}^k \,;\, \theta) \,. \tag{3}$$

When PG and PQT objectives are combined, their respective gradients can simply be added together to arrive at the joint gradient. In the joint setting the priority queue component has a stabilizing affect and helps reduce catastrophic forgetting in the policy. This approach bears some similarity to the RL approaches adopted by Google's Neural Machine Translation (Wu et al., 2016) and Neural Symbolic Machines (Liang et al., 2017).

**Entropy exploration.** We also regularize the policy by adding an entropy term which aims to increase the uncertainty of the model and encourage exploration. This prevents the policy from assigning too much probability mass to any particular sequence, thereby encouraging more diversity among sampled programs. This regularizer has been prescribed initially by Williams & Peng (1991) and more recently adopted by Mnih et al. (2016); Nachum et al. (2017b). We use the entropy regularizer for both PG and PQT.

The most general form of the objective can be expressed as the sum of all of these components into one quantity. We assign different scalar weights to the PG, PQT, and entropy terms, and the gradient of the overall objective is expressed as,

$$\lambda_{\text{ER}} \nabla_\theta O_{\text{ER}}(\theta) + \frac{\lambda_{\text{TOPK}}}{K} \sum_{k=1}^{K} \nabla_\theta \log \pi(\tilde{a}_{1:\tilde{T}_k}^k \,;\, \theta) + \frac{\lambda_{\text{ENT}}}{N} \sum_{i=1}^{N} \sum_{t=1}^{T_i} \nabla_\theta H[\pi(A \mid a_{1:t-1}^i \,;\, \theta)] \,, \tag{4}$$

where entropy $H[p(X)] = -\sum_{x \in X} p(x) \log p(x)$. The optimization goal is the maximize (4). Any specific term can be remove by settings its corresponding $\lambda$ to 0. When we train vanilla PG, $\lambda_{\text{TOPK}} = 0$, and when we train with PQT, $\lambda_{\text{ER}} = 0$.

**Distributed training.** To speed up the training of the program synthesizers, we also make use of an asynchronous distributed setup, where a parameter server stores the shared model parameters for a number of synthesizer replicas. Each synthesizer replica samples a batch of episodes from its local copy of the policy and computes the gradients. Then, the gradients are sent to the parameter server, which asynchronously updates the shared parameters (Mnih et al., 2016). The replicas periodically update their local policy with up-to-date parameters from the parameter server. Also, to make the implementation of distributed PQT simple, each replica has its own priority queue of size $K$. We use 32 replicas for all of the experiments.

## 4 EXPERIMENTS

We assess the effectiveness of our program synthesis setup by trying to discover BF programs. In what follows, we first describe the BF programming language. Then we discuss the tasks that were considered, and we present our experimental protocol and results.

## 4.1 THE BF PROGRAMMING LANGUAGE

BF is a minimalist Turing complete language consisting of only 8 low-level operations, each represented by a char from $\boxed{\texttt{+-<>[].,}}$. See Table 1 for operation descriptions. Operations are executed from left to right and square brackets enable looping, which is the only control flow available in the language. BF programs operate on a memory tape and internally manipulate a data pointer. The data pointer is unbounded in the positive direction but is not permitted to be negative. Memory values are not accessed by address, but relatively by shifting the data pointer left or right, akin to a Turing machine. Likewise, to change a value in memory, only increment and decrement operations are available (overflow and underflow is allowed). We include an execution demo in Appendix 6.1 to further aid understanding of the language.

BF programs are able to read from an input stream and write to an output stream (one int at a time). This is how inputs and outputs are passed into and out of a BF program in our tasks. In our implementation $\boxed{\texttt{,}}$ will write zeros once the end of the input stream is reached, and many synthesized programs make use of this feature to zero out memory.

Memory values are integers, typically 1 byte and so they are often interpreted as chars for string tasks. In our BF implementation the interpreter is given a task-dependent base $B$ so that each int is in $\mathbb{Z}_B$, *i.e.,* the set of integers modulo base $B$. By default $B = 256$, unless otherwise specified.

Table 1: The eight commands in the BF programming language (c.f. `https://esolangs.org/wiki/Brainfuck`).

| Command | Description |
|---------|-------------|
| > | Move the data pointer to the right |
| < | Move the data pointer to the left |
| + | Increment the value at the current memory position |
| − | Decrement the value at the current memory position |
| . | Output the value at the current memory position |
| , | Get next value from the input stream and write it to the current memory position |
| [ | Jump past the matching ] if the cell under the pointer is 0 |
| ] | Jump back to the matching [ if the cell under the pointer is nonzero |

In our main experiments programs are fixed length, and most characters end up being useless no-ops. There are many ways to make no-ops in BF, and so it is very easy to pad out programs. For example, the move left operation $\boxed{\texttt{<}}$ when the data pointer is at the leftmost position is a no-op. Unmatched braces when strict mode is off are also no-ops. Putting opposite operations together, like $\boxed{\texttt{+-}}$ or $\boxed{\texttt{<>}}$, work as no-op pairs.

Notice that there is only one type of syntax error in BF: unmatched braces. BF's extremely simple syntax is another advantage for program synthesis. For languages with more complex syntax, synthesizing at the character level would be very difficult due to the fact that most programs will not run or compile, thus the reward landscape is even sparser. We gave our BF interpreter a flag which turns off "strict mode" so that unmatched braces are just ignored. We found that turning off strict mode makes synthesis easier.

## 4.2 REWARDS

To evaluate a given program under a task, the code is executed on one or many test cases sampled from the task (separate execution for each test input). Each test case is scored based on the program's output and the scores are summed to compute the final reward for the program.

More formally, let $\mathcal{T}$ be the task we want to synthesize code for, and let $P$ be a candidate program. We treat $P$ as a function of input $I$ so that output $Q = P(I)$. Inputs and outputs are lists of integers (in base $B$). In principle for any NP task a polynomial time reward function can be computed. A trivial reward function would assign $1.0$ to correct outputs and $0.0$ to everything else. However, such 0/1 reward functions are extremely sparse and non-smooth (*i.e.,* a code string with reward $1.0$ may

be completely surrounded by strings of reward $0.0$ in edit-distance space). In practice a somewhat graded reward function must be used in order for learning to take place.

The following formulation presents a possible way to compute rewards for these tasks or other similar tasks, but our method does not depend on any particular form of the reward (as long as it is not too sparse). Because all of the tasks we chose in our experiments are in the polynomial time class, the easiest way for us to score program outputs is just to directly compare to the correct output $Q^*$ for a given input $I$. We use a continuous comparison metric between $Q$ and $Q^*$ to reduce reward sparsity.

To evaluate a given program $P$ under task $\mathcal{T}$, a set of test cases is sampled $\{(I_1, Q_1^*), ..., (I_n, Q_n^*)\}$ from $\mathcal{T}$. We leave open the option for $\mathcal{T}$ to produce static test cases, *i.e.,* the same test cases are provided each time, or stochastic test cases drawn from a distribution each time. In practice, we find that static test cases are helpful for learning most tasks (see Section 4.3).

For simplicity we define a standardized scoring function $S(Q, Q^*)$ for all our tasks and test cases (see Appendix 6.2 for details). Total reward for the program is $\mathcal{R}^{tot} = \zeta \sum_{i=1}^{n} S(P(I_i), Q_i^*)$ where $\zeta$ is a constant scaling factor, which can differ across tasks. We use $\zeta$ to keep rewards approximately in the range $[-1, 1]$. $\mathcal{R}^{tot}$ is given to the agent as terminal reward. When generating variable length programs we give preference to shorter programs by adding a program length bonus to the total reward: $\mathcal{R}^{tot} + 1 - |P|/\text{MaxProgramLength}$.

If the BF interpreter is running in strict mode and there is a syntax error (*i.e.,* unmatched braces) we assign the program a small negative reward. We also assign negative reward to programs which exceed 5000 execution steps to prevent infinite loops. Note that $\mathcal{R}^{tot}$ is the terminal reward for the agent.

## 4.3 EXPERIMENTAL SETUP

We assess the effectiveness of our priority queue training method against the following baselines:

- Genetic algorithm (GA) implemented based on Becker & Gottschlich (2017).[2] See Appendix 6.3 for more details regarding our implementation.
- Policy gradient (PG) as described in Section where $\lambda_{\text{TOPK}} = 0$.
- Policy gradient (PG) combined with priority queue training (PQT), where $\lambda_{\text{TOPK}} > 0$.

We are operating under the RL paradigm, and so there is no test/evaluation phase. We use a set of benchmark coding tasks (listed in Appendix 6.4) to compare these methods, where different models are trained on each task. Simply the best program found during training is used as the final program for that task.

In order to tune each method, we propose to carry out experiments in two phases. First, the hyperparameters will be tuned on a subset of the tasks (*reverse* and *remove-char*). Next, the best hyperparameters will be fixed, and each synthesis method will be trained on all tasks. To compare performance between the synthesis methods, we measure the success rate of each method after a predetermined number of executed programs. More details will be described in Section 4.4.

We tune the hyperparameters of the synthesis methods on *reverse* and *remove-char* tasks. We use grid search to find the best hyperparameters in a given set of possible values. The tuning space is as follows. For PG, learning rate $\in \{10^{-5}, 10^{-4}, 10^{-3}\}$ and entropy regularizer $\in \{0.005, 0.01, 0.05, 0.10\}$. For PQT, learning rate and entropy regularizer are searched in the same spaces, and we also allow the entropy regularizer to be 0; PQT loss multiplier ($\lambda_{\text{TOPK}}$ from Equation 4) is searched in $\{1.0, 10.0, 50.0, 200.0\}$. For GA, population size $\in \{10, 25, 50, 100, 500\}$, crossover rate $\in \{0.2, 0.5, 0.7, 0.9, 0.95\}$ and mutation rate $\in \{0.01, 0.03, 0.05, 0.1, 0.15\}$.

For PQT we set $K = 10$ (maximum size of the priority queue) in all experiments. In early experiments we found 10 is a nice compromise between a very small queue which is too easy for the RNN to memorize, and a large queue which can dilute the training pool with bad programs.

The best hyperparameters found by grid search for each synthesis method are:

---

[2]Author's implementation available at `https://github.com/primaryobjects/AI-Programmer`

- **PG**: entropy regularizer = 0.05, learning rate = $10^{-4}$.
- **PG+PQT**: entropy regularizer = 0.01, learning rate = $10^{-4}$, $\lambda_{\text{TOPK}}$ = 50.0.
- **PQT**: entropy regularizer = 0.01, learning rate = $10^{-4}$, $\lambda_{\text{TOPK}}$ = 200.0.
- **GA**: population size = 100, crossover rate = 0.95, mutation rate = 0.15.

**Other model and environment choices.** For PG and PQT methods we use the following architecture: a 2-layer LSTM RNN (Hochreiter & Schmidhuber, 1997) with 35 units in each layer. We jointly train embeddings for the program symbols of size 10. The outputs of the top LSTM layer are passed through a linear layer with 8 or 9 outputs (8 BF ops plus an optional EOS token) which are used as logits for the softmax policy. We train on minibatches of size 64, and use 32 asynchronous training replicas. Additional hyperparameter values: gradient norm clipping threshold is set at 50, parameter initialization factor[3] is set at 0.5, RMSProp is used as the optimizer, and decay for the exponential moving average baseline is set at 0.99.

We explored a number of strategies for making test cases. For the *reverse* task we tried:

1. One random test case of random length.
2. Five random test cases of random length.
3. Five random test cases of lengths 1 through 5.
4. Five static test cases of lengths 1 through 5.

However, solutions were only found when we used static test cases (option 4).

In the experiments below, all programs in each task are evaluated on the same test cases. The test inputs are randomly generated with a fixed seed before any training happens. By default each task has 16 test cases, with a few exceptions noted in Appendix 6.4. For the two tuning tasks we continue to use a small number of hand crafted test cases.

A potential problem with using the test cases for code synthesis is that the synthesized code can overfit, *i.e.*, the code can contain hard-coded solutions for test inputs. In the experiments below we also run synthesized code on a large set of held-out eval test cases. These eval test cases are also randomly generated with a fixed seed, and the total number of test cases (train and eval) for each task is 1000. Success rates on training test cases and all test cases are reported in Table 3. We do manual inspection of code solutions in Table 4 to identify overfitting programs, which are highlighted in red.

## 4.4 RESULTS

In the following, we show success rates on tuning tasks and held-out tasks for all algorithms. Again, our metric to compare these methods is the success rate of each method at finding the correct program after a predetermined maximum number of executed programs. For all tasks, we ran training 25 times independently to estimate success rates. A training run is successful if a program is found which solves all the test cases for the task. Training is stopped when the maximum number of programs executed (NPE) is reached, and the run is considered a failure. For tuning, we use maximum NPE of 5M. For evaluation we use a maximum NPE of 20M.

Our genetic algorithm is best suited for generating programs of fixed length, meaning all code strings considered are of some fixed length preset by the experimenter. In all the experiments presented below, the EOS token is disabled for all algorithms, so that there are 8 possible code characters. Program length is 100 characters for all experiments, and the search space size is $8^{100} \approx 10^{90}$.

In Table 2, we report the success rates from tuning of the genetic algorithm, policy gradient, priority queue training, and policy gradient with priority queue training. The results for these tuning tasks are different from the same tasks in Table 3, due to the smaller NPE used in tuning, and the fact that we tune on a different set of hand-selected test cases. There is also sensitivity of each synthesis method to initialization, sampling noise, and asynchronous weight updates which accounts for differences between multiple runs of the same tasks.

---

[3]Parameter initialization factor is the factor argument for the TensorFlow variable initializer `tf.contrib.layers.variance_scaling_initializer`.

Table 2: Number of successes (out of 25) of synthesis methods on tuning tasks when Maximum Number of Programs Executed (NPE) is 5M.

| Task | GA | PG | PQT | PG+PQT |
|---|---|---|---|---|
| reverse | 12 | 2 | 20 | 21 |
| remove-char | 12 | 5 | 5 | 1 |

In Table 3, we report the success rates of the same algorithms plus uniform random search. We include success rates for training and eval test cases. We also do an aggregate comparison between columns by taking the average at the bottom. As can be seen from the table, PQT is clearly better than PG and GA according to training and eval averages. PG+PQT is on par with PQT alone. The eval success rates are lower than the training success rates in many cases due to overfitting programs.

Table 3: Number of successes (out of 25) of synthesis methods on all tasks when the maximum number of programs executed (max NPE) is 20M. In each cell we report two numbers separated by a forward slash. The number of successes on training test cases is first, and the number of successes on held-out eval test cases is second. For many task-method combinations no program was found which satisfies the training cases, and we mark these cells with just a dash.

| Task | Uniform | GA | PG | PQT | PG+PQT |
|---|---|---|---|---|---|
| reverse | 2 / 2 | 15 / 15 | 3 / 2 | 20 / 20 | 17 / 17 |
| remove-char | - | 21 / 0 | 2 / 0 | 18 / 0 | 10 / 0 |
| count-char | - | 4 / 4 | - | - | - |
| add | 2 / 2 | 19 / 18 | 6 / 6 | 25 / 25 | 25 / 25 |
| bool-logic | - | 19 / 18 | 19 / 19 | 15 / 15 | 17 / 17 |
| print-hello | - | 12 / 16 | - | 25 / 25 | 25 / 25 |
| echo-twice | - | 11 / 2 | - | 3 / 1 | 5 / 1 |
| echo-thrice | - | - | - | - | - |
| copy-reverse | - | - | - | - | - |
| zero-cascade | - | 1 / 0 | - | 21 / 1 | 22 / 0 |
| cascade | - | - | - | 11 / 1 | 9 / 0 |
| shift-left | 13 / 0 | 25 / 2 | 13 / 2 | 25 / 0 | 25 / 0 |
| shift-right | - | 3 / 0 | - | - | - |
| riffle | - | - | - | - | - |
| unriffle | - | 4 / 0 | - | 8 / 0 | 8 / 0 |
| middle-char | - | 1 / 0 | - | 17 / 0 | 22 / 0 |
| remove-last | 1 / 1 | 19 / 13 | - | 25 / 9 | 25 / 5 |
| remove-last-two | - | 2 / 0 | - | 25 / 9 | 25 / 8 |
| echo-alternating | - | 4 / 0 | - | 25 / 0 | 25 / 0 |
| echo-half | - | - | - | - | 1 / 0 |
| length | 22 / 10 | 25 / 5 | 17 / 2 | 25 / 12 | 25 / 10 |
| echo-second-seq | 25 / 10 | 25 / 5 | 25 / 9 | 25 / 8 | 25 / 7 |
| echo-nth-seq | - | 13 / 13 | - | 24 / 23 | 25 / 25 |
| substring | - | - | - | - | - |
| divide-2 | - | - | - | - | - |
| dedup | - | - | - | - | - |
| Average | 2.5 / 1.0 | 8.6 / 4.3 | 3.3 / 1.5 | **13.0 / 5.7** | 12.9 / 5.4 |

## 4.5 SAMPLE PROGRAMS

In this section we have our method generate shortest possible code string. Code shortening, sometimes called *code golf*, is a common competitive exercise among human BF programmers. We use PG+PQT to generate programs of variable length (RNN must output EOS token) with a length bonus in the reward to encourage code simplification (see 4.2). We train each task just once, but with a much larger maximum NPE of 500M. We do not stop training early, so that the agent can iterate on known solutions. We find that alternative hyperparameters work better for code shortening, with $\lambda_{\text{ENT}} = 0.05$ and $\lambda_{\text{TOPK}} = 0.5$.

In Table 4 we show simplified programs for coding tasks where a solution was found.

| Task | Code |
|------|------|
| reverse | `,[>,]+[,<.]` |
| remove-char | `,-[+.,-]+[,.]` |
| count-char | `,[-[>]>+<<<<,]>.` |
| add | `,[+>,<<->],<.,.` |
| bool-logic | `,+>,<[,>],<+<.` |
| print-hello | `++++++++.--.+++++++..+++.` |
| zero-cascade | `,.,[.>.-<,[[[.+,>+[-.>]..<]>+<<]>+<<]]` |
| cascade | `,[.,.[.,.[..,[....,[.....,[.>]<]].]]` |
| shift-left | `,>,[.,]<.>.` |
| shift-right | `,[>,]<.,<<<<<.[>.]` |
| unriffle | `-[,>,[.,>,]<[>,]<.]` |
| remove-last | `,>,[<.>>,].` |
| remove-last-two | `>,<,>>,[<.,[<.[>]],].` |
| echo-alternating | `,[.,>,]<<<<.[>.]` |
| length | `,[>+<,]>.` |
| echo-second-seq | `,[,]-[,.]` |
| echo-nth-seq | `,-[->-[,]<]-[,.]` |

Table 4: Synthesized BF programs for solved tasks. All programs were discovered by the agent as-is, and no code characters were removed or altered in anyway. Notice that some programs overfit their tasks. For example, *cascade* is correct only for up to length 6 (provided test cases are no longer than this). We highlighted in red all the tasks with code solutions that overfit, which we determined by manual inspection.

## 5 DISCUSSION

In this paper, we considered the task of learning to synthesize programs for problems where a reward function is defined. We use an RNN trained with our priority queue training method. We experimented with BF, a simple Turing-complete programming language, and compared our method against a genetic algorithm baseline. Our experimental results showed that our method is more stable than vanilla policy gradient or a genetic algorithm.

That PQT works as a standalone search algorithm is surprising, and future work is needed in order to better explain it. We can speculate that it is implementing a simple hill climbing algorithm where the buffer stores the best known samples, thereby saving progress, while the RNN functions as an exploration mechanism. Even more surprising is that this algorithm is able to bootstrap itself to a solution starting from an empty buffer and a randomly initialized RNN. We believe that our coding environment complements the PQT algorithm, since finding code with non-zero reward through purely random search is feasible.

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

## 6  APPENDIX

### 6.1  BF EXECUTION DEMO

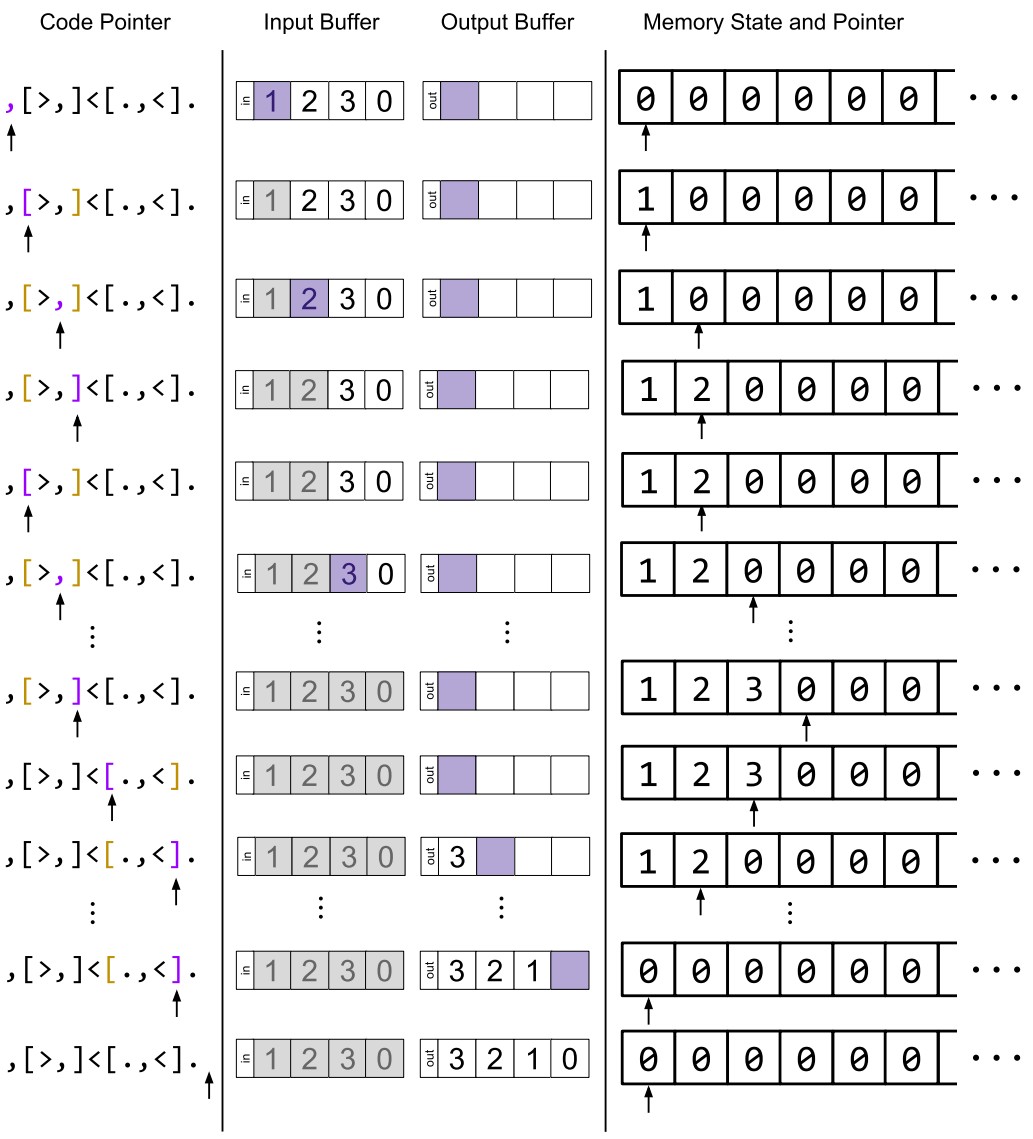

Figure 2: In the following figure we step through a BF program that reverses a given list. The target list is loaded into the input buffer, and the programs output will be written to the output buffer. Each row depicts the state of the program and memory before executing that step. Purple indicates that some action will be taken when the current step is executed. We skip some steps which are easy to infer. Vertical ellipses indicate the continuation of a loop until its completion.

### 6.2  TEST CASE SCORING

Our implementation of $S(Q, Q^*)$ is computed from a non-symmetric distance function $d(Q, Q^*)$ which is an extension of Hamming distance $H(\text{list}_1, \text{list}_2)$ for non-equal length lists (note that arguments to $H$ must be the same length). Hamming distance provides additional information, rather than just saying values in $Q$ and $Q^*$ are equal or not equal. Further since BF operates on values only

through increment and decrement operations this notation of distance is very useful for conveying to the agent information about how many inc or dec ops to use in various places. This serves to make the reward space smoother and less sparse.

We want a distance of 0 to result in the maximum score and a large distance to result in a small or negative score. Thus we define:

$$S(Q, Q^*) = d(\emptyset, Q^*) - d(Q, Q^*)$$

where $\emptyset$ is the empty list, $l = |Q|$, $l^* = |Q^*|$, and $B$ is the integer base (number of possible ints at each position). We define our distance function:

$$d(Q, Q^*) = \begin{cases} H(Q, Q^*_{1:l}) + B \cdot (l^* - l) & \text{if } l \leq l^* \\ H(Q_{1:l^*}, Q^*) + B \cdot (l - l^*) & \text{otherwise} \end{cases}$$

Essentially $d(Q, Q^*)$ adds maximum char distance to the Hamming distance for each missing position or each extra position, depending on whether $Q$ is shorter or longer than $Q^*$ respectively. $S(Q, Q^*)$ subtracts the distance $d(Q, Q^*)$ from the distance of an empty list to $Q^*$, which is equal to $B|Q^*|$.

## 6.3 GENETIC ALGORITHM

A genetic algorithm (GA) simulates sexual reproduction in a population of genomes in order to optimize a fitness function. To synthesize BF programs, we let a genome be one code string. The GA is initialized with a population of randomly chosen code strings. For each iteration a new population of children is sampled from the existing population. Each new population is called a generation.

GA samples a new population from the current population with 3 steps: 1) parent selection, 2) mating, and 3) mutation. Many algorithms have been developed for each of these steps which have varying effects on the GA as a whole. We describe our algorithm for each step:

1. **Parent selection:** Randomly sample a set of parents from the population. We use roulette selection (a.k.a. fitness proportionate selection) where parents are chosen with probability proportional to their fitness.

2. **Mating:** Choose pairs of parents and perform an operation resulting in two children to replace the parents. We use single point crossover where a position in the genome (code string) for the first parent is sampled uniformly and the two parents' genetic material is swapped after that point to create two new children. Crossover is performed with probability $p_{crossover}$.

3. **Mutation:** With some probability make small random modifications to each child. We use the *primaryobjects* mutation function. This function iterates through each code token and with probability $p_{mutate}$ chooses among 4 possible mutation operations to apply: insert random token, replace with random token, delete token, shift and rotate either left or right. When inserting the last token is removed, and when deleting a random token is added at the end so that none of the mutation operations change the number of tokens in the list.

We tune $p_{mutate}$, $p_{crossover}$, and the population size (see *Experimental Setup*). For each task, the genome size is set to the maximum code length, since GA operates on fixed length genomes.

## 6.4 CODING TASKS

Table 5: List of coding tasks with descriptions. For any task, a BF program takes a single list as input, and outputs a list. Unless otherwise state base $B = 256$, and each value can be considered as an ASCII character.

| Task | Description |
|------|-------------|
| reverse | Return input in reverse order. |
| remove-char | Remove all `1`s from the input and return the result. |
| count-char | Count number of occurrences of `1` in the input. |
| add | Return the sum (modulo 256) of two input numbers. There are 9 hand picked test cases. |
| bool-logic | Read in 3 bools $x, y, z$ and return $f(z, y, z) = x\bar{z} + \bar{y}\bar{z} + \bar{x}yz$. Base $B = 2$. There are 8 test cases, one for each of the possible 3 bit combinations. |
| print-hello | Return 'HELLO'. Base $B = 27$, where 'A' = 1, ..., 'Z' = 26, and EOS = 0. There is one test case: the target string. |
| echo-twice | Return the input repeated twice. |
| echo-thrice | Return the input repeated three times consecutively. |
| copy-reverse | Return the input, followed by the input reversed, and followed by the original input. |
| zero-cascade | For all input values, return the $i^{\text{th}}$ value followed by $i$ 0s. |
| cascade | For all input values, return the $i^{\text{th}}$ value $i$ times. |
| shift-left | Circular shift the input left, so that the first value is last. |
| shift-right | Circular shift the input right, so that the last value is first. |
| riffle | For input of length $N$, return the $(n-1)^{\text{th}}$ input, then the $0^{\text{th}}$, then $(n-2), 1, (n-3), 2, ...$ |
| unriffle | Inverse of the *riffle* function. |
| middle-char | For input of length $N$, return the value at position $floor(N/2)$. |
| remove-last | Remove the last character from the list. |
| remove-last-two | Remove the last two characters from the list. |
| echo-alternating | Return every even indexed value, followed by every odd indexed value. |
| echo-half | Return the first half of the input. |
| length | Return the length of the list. |
| echo-nth-seq | For $M$ input sequences each seperated by a `0`, return the $n^{\text{th}}$ sequence, where $n$ is given as the first value in the input. |
| substring | Return a sub-range of the input list, given a starting index $i$ and length $l$. |
| divide-2 | Return input value divided by two (integer division). |
| dedup | Return input list, in which all duplicate adjacent values removed. |

