# OpenReview forum: "Code Synthesis with Priority Queue Training"
_ICLR.cc/2018/Conference — Reject_

### Official Review · AnonReviewer2 · 2017-11-25
**Interesting PQT approach using top-K highest reward programs**

**Rating:** 6
**Confidence:** 4

**Review:**

This paper presents an algorithm called Priority Queue Training (PQT) for
program synthesis using an RNN where the RNN is trained in presence of a
reward signal over the desired program outputs. The RNN learns a policy
that generates a sequence of characters in BF conditioned on a prefix of characters.
The key idea in PQT is to maintain a buffer of top-K programs at each
gradient update step, and use them to perform additional supervised learning
of the policy to guide the policy towards generating higher reward programs.
PQT is compared against genetic algorithm (GA) and policy gradient (PG) based
approaches on a set of BF benchmarks, where PQT is able to achieve the most
number of average successes out of 25 runs.

Unlike previous synthesis approaches that use supervision in terms of ground
truth programs or outputs, the presented technique only requires a reward
function, which is much more general. It is impressive to see that a simple
technique of using top-K programs to provide additional supervision during
training can outperform strong GA and PG baselines.

It seems that the PQT approach is dependent on being able to come up with a
reasonably good initial policy \pi such that the top-K programs in the priority
queue are reasonable, otherwise the supervised signal might make the RNN policy
worse. How many iterations are needed for PQT to come up with the target programs?
It would be interesting to see the curve that plots the statistics about the
rewards of the top-K programs in the priority queue over the number of iterations.

Is there also some assumption being made here in terms of the class of programs
that can be learnt using the current scoring function? For example, can the if/then
conditional tasks from the AI Programmer paper be learnt using the presented
scoring function?

I have several questions regarding the evaluation and comparison metrics.

For the tuning task (remove-char), GA seems to achieve a larger number of
successes (12) compared to PQT (5) and PG+PQT (1). Is it only because the
NPE is 5M or is there some other reasons for the large gap?

The criterion for success in the evaluation is not a standard one for program
synthesis. Ideally, the success criterion should be whether a technique is able
to find any program that is consistent with the examples. Observing the results
in Table 3, it looks like GA is able to synthesize a program for 3 more benchmarks
(count-char, copy-reverse, and substring) whereas PQT is only able to solve one
more benchmark (cascade) than GA (on which PQT instead seems to overfit).
Is there some insights into how the PQT approach can be made to solve those
tasks that GA is able to solve?

How do the numbers in Table 3 look like when the NPE is 5M and when NPE is
larger say 100M?

There are many learnt programs in Table 4 that do not seem to generalize to new
inputs. How is a learnt program checked to be correct in the results reported
in Table 3? It seems the current criterion is just to check for correctness on
a set of 5 to 20 predefined static test cases used during training. For every
benchmark, it would be good to separately construct a set of held out test cases
(possibly of larger lengths) to evaluate the generalization correctness of the
learnt programs. How would the numbers in Table 3 look with such a correctness
criterion of evaluating on a held-out set?

Are there some insights regarding why programs such as divide-2, dedup or echo-thrice
can not be learnt by PQT or any other approach? The GA approach in the AI programmer
paper is able to learn multiply-2 and multiply-3 programs that seems comparable to
the complexity of divide-2 task. Can PQT learn multiply-3 program as well?

---

> ### Author Response · Authors · 2017-12-22
> **Response**
>
> Thank you for your review. We address your key concerns:
>
> The reviewer argues that PQT depends on a good initial policy. We agree that PQT depends on the initial policy having appropriate bias so that it is able to sample programs with non-zero reward, but genetic algorithms and policy gradient methods require this initial bias as well. All the search methods we consider in our paper need to be able to bootstrap from no information, and rely on stumbling upon code strings with non-trivial reward in the initial stages. We also intentionally designed the our coding environment such that these search methods are able to bootstrap themselves, by making it so that all possible program strings can be executed and receive reward.
>
> The reviewer is concerned that strong assumptions are made with the current scoring function. We agree that our scoring function does determine what coding tasks are solvable in our setup. We see this as an unavoidable drawback to the problem of code synthesis under a reward function. A truly unbiased reward function would be too sparse and make the search infeasible. In shaping our reward function, there is no avoiding building in bias for certain types of problems. We hope that future work can be done in removing the need to shape the reward function.
>
> The reviewer argues that GA appears to solve more tasks than PQT. We disagree, and as we stated in our response to AnonReviewer1, we do not feel that this difference (in which tasks are solved) between GA and PQT is statistically significant. Since these results have high variance, that motivated our decision to compare success rates instead.
>
> Regarding other concerns:
>
> The reviewer asks why PQT does so poorly in tuning on the remove-char task, while doing much better in eval. As the reviewer points out, the lower NPE is one reason. Another reason is that the test cases we use for the remove-char task in tuning are not the same as the ones used in eval. We update the paper to state these differences.
>
> The reviewer asks “How do the numbers in Table 3 look like when the NPE is 5M and when NPE is larger say 100M?” We did some initial experiments with NPE of 5M, and did not observe any significant difference over 20M. We imagine that lower NPE would reduce success rates overall. We believe that 20M is approaching the upper bound on what is reasonable in terms of compute and time. For reproducibility, we did not want to use larger NPEs.
>
> The reviewer asks "How is a learnt program checked to be correct in the results reported
> in Table 3?" We do not check generalization of programs in Table 3. AnonReviewer1 also asks why we do not have held out test cases to test generalization, and we admit that to be an oversight in our experimental setup. We update Table 4 in the paper to highlight which tasks were observed to overfit via manual inspection.
>
> The reviewer asks if PQT can learn multiply-3, which was a task in the AI programmer paper. We decided that multiply-N tasks were too easy, but we did not verify this.
> For example, here are solutions to multiply-2, multiply-3, and multiply-4:
> ,[->++<]>.
> ,[->+++<]>.
> ,[->++++<]>.
> We feel these are fairly short and contain just a single loop. We already included a few simple tasks and didn't want to add more. We also did not include the if/then task for the same reason.

---

### Official Review · AnonReviewer1 · 2017-11-27
**Focus on the success**

**Rating:** 5
**Confidence:** 3

**Review:**

This paper focuses on using RNNs to generate straightline computer programs (ie. code strings) using reinforcement learning.  The basic setup assumes a setting where we do not have access to input/output samples, but instead only have access to a separate reward function for each desired program that indicates how close a predicted program is to the correct one.  This reward function is used to train a separate RNN for each desired program.

The general space of generating straight-line programs of this form has been explored before, and their main contribution is the use of a priorty queue of highest scoring programs during training.  This queue contains the highest scoring programs which have been observed at any point in the training so far, and they consider two different objectives:  (1) the standard policy-gradient objective which tries to maximize the expected reward and (2) a supervised learning objective which tries to maximize the average probability of the top-K samples.  They show that this priority queue algorithm significantly improves the stability of the resulting synthesis procedure such that when synthesis succeeds at all, it succeeds for most of the random seeds used.

This is a nice result, but I did not feel as though their algorithm was sufficently different from the algorithm used by Liang et. al. 2017.  In Liang et. al. they keep around the best observed program for each input sample.  They argue that their work is different from Liang et. al. because they show that they can learn effectively using only objective (2) while completely dropping objective (1).  However I'm quite worried these results only apply in very specific setups.  It seems that if the policy gradient objective is not used, and there are not K different programs which generate the correct output, then the Top-K objective alone will encourage the model to continue to put equal probability on the programs in the Top-K which do not generate an incorrect output.

I also found the setup itself to be poorly motivated.  I was not able to imagine a reasonable setting where we would have access to a reward function of this form without input/output examples.  The paper did not provide any such examples, and in their experiments they implement the proposed reward function by assuming access to a set of input/output examples.  I feel as though the restriction to the reward function in this case makes the problem uncessarily hard, and does not represent an important use-case.

In addition I had the following more minor concerns:

1.  At the end of section 4.3 the paper is inconsistent about whether the test cases are randomly generated or hand picked, and whether they use 5 test cases for all problems, or sometimes up to 20 test cases.  If they are hand picked (and the number of test cases is hand chosen for each problem), then how dependant are the results on an appropriate choice of test cases?

2.  They argue that they don't need to separate train and test, but I think it is important to be sure that the generated programs work on test cases that are not a part of the reward function.  They say that "almost always" the synthesizer does not overfit, but I would have liked them to be clear about whether their reported results include any cases of overfitting (i.e. did they ensure they the final generate program always generalized)?

3.  It is worth noting that while their technique succeeds much more consistently than the baseline genetic algorithm, the genetic algorithm actually succeeds at least once, on more tasks (19 vs. 17).  The success rate is probably a good indicator of whether the technique will scale to more complex problems, but I would have prefered to see this in the results, rather than just hoping it will be true (i.e. by including my complicated problems where the genetic algorithm never succeeds).

---

> ### Author Response · Authors · 2017-12-22
> **Response**
>
> Thank you for your review. We address your key concerns:
>
> Regarding the novelty of our method: After considering the reviewer's concerns, we are happy to soften the claim on novelty. We have updated the paper to give more credit to prior work, such as Liang et al 2017. However, we believe there are some key differences between our method and Liang et al 2017 which make PQT interesting. Mainly, our method does not use beam search, and we do not take any input into the RNN. Additionally, even though they are similar, it is important that our empirical evidence suggests that topk training without reinforcement learning is good enough. This further simplifies program synthesis and potentially has implications to other areas in reinforcement learning.
>
> The reviewer is concerned that our results only apply in very specific setups, because "the Top-K objective alone will encourage the model to continue to put equal probability on the programs in the Top-K [buffer] which do not generate [a correct] output." We agree that putting equal weight on all top-K solutions may be problematic in some situations, but we also believe there are improvements that can be made to PQT which remove the issue. For instance, one could imagine sampling from the top-K buffer with probability proportional to reward (or some transformation on rewards to make all sampling weights positive). We show that PQT as presented in the paper is a viable method, and leave improvements to future work.
>
> Regarding the problem setup is being too restrictive: We address this in our response to AnonReviewer3. Though we agree that hiding input/output examples from the code synthesizer makes synthesis harder, we feel that the problem setup becomes more elegant, as it reduces to a search problem over a reward function. We would like to stress that our experimental setup is motivated by our goal to simplify the problem in order to isolate aspects of code synthesis that are fundamentally hard, and to show that general methods are viable for solving this problem.
>
> Regarding other concerns:
>
> 1) This reviewer's confusion is due to a mistake in the paper. We fix the language and hope that clears up confusion around choice of test cases.
>
> 2) The reviewer comments “They argue that they don't need to separate train and test...”
> We agree that having held out eval cases would have been better, and we admit that to be an oversight in our design of the experimental setup. We update Table 4 in the paper to highlight which tasks were observed to overfit via manual inspection. We also note the reviewer's confusion around the sentence "Almost always the synthesizer does not overfit." We agree that this language was imprecise and unhelpful to the reader, and we remove it from the paper.
>
> 3) Regarding the metric for success: We chose to compare success rate across tasks, rather than absolute number of tasks solved, because the latter has higher variance. Two of the tasks which GA solves over PQT, copy-reverse and substring, have a success of 1 out of 25. We feel these results are not statistically significant enough to draw conclusions, since it is always possible that PQT would also achieve a small amount of successes on these tasks given more runs. As for the cases where GA clearly solves a task over PQT (and vise-versa), we also feel these differences are not significant enough to draw conclusions, as this only happens once for each method (count-char for GA and cascade for PQT).
>
> Additionally the reviewer comments “The success rate is probably a good indicator of whether the technique will scale to more complex problems, but I would have prefered to see this in the results, rather than just hoping it will be true (i.e. by including my complicated problems where the genetic algorithm never succeeds).” We agree that there is uncertainty around whether success rate is a good indicator of whether the technique will scale to more complex problems. However, we were not able to come up with adequate way to measure that. Including complicated problems where the genetic algorithm never succeeds, as the reviewer suggests, implies having to find tasks where specifically GA fails while PQT succeedes. We feel this is the same as picking data which supports our conclusion, and would not be a scientific way to choose tasks. We took care to select the tasks in Table 3 before knowing how GA and PQT will perform on them.

---

> > ### Author Response · Authors · 2017-12-22
> > **Reward functions without known outputs to test cases**
> >
> > Regarding the reviewer's comment, "I was not able to imagine a reasonable setting where we would have access to a reward function of this form without input/output examples." We believe that for any coding task where there exists an algorithm for checking a solution, that algorithm can be used to compute reward on a test input. We take the reviewer's statement, "reward function of this form," to mean a shaped reward function, i.e. where there is some quantitative notion of "goodness" for incorrect code outputs, so that reward gets larger as the output gets closer to the correct output. We provide two such examples:
> >
> > Example 1: Consider the task of synthesizing code for the well known traveling salesman problem (TSP). One could naively construct a shaped reward function that takes only a set of test inputs and a candidate code string. Since the goal of TSP is to produce the shortest path given a set of vertices, the reward for each test input (set of vertices) can just be the negative of the path length returned by the candidate code, with penalties for invalid paths (paths that do not hit every city exactly once). The total reward is the sum of rewards for each test input. Though it is likely not feasible to synthesize code for the TSP problem in BF, our experimental setup is general enough to allow non-deterministic polynomial time coding problems to be considered.
> >
> > Example 2: Consider the task of synthesizing code to sort lists in ascending order. A simple way to compute shaped reward given a test input and an output emitted by a candidate program, is to count the number of adjacent pairs which are in ascending order in the output list, and subtract off penalties for elements which were not contained in the input or are missing from the output.

---

> > > ### Comment · AnonReviewer1 · 2018-01-03
> > > **RE: Response**
> > >
> > > Thanks for your response, and your fixes to the paper. I still have a couple of concerns, however. (1) Why did you only update the results in Table 4 with manual inspection, and not the results in Table 3?  Presumably the number of different programs generated is not so huge that manual inspection is unreasonable?  (2) Unfortunately, I don't find your reward function examples particularly convincing.  I agree, that there are certain particular problems where it may be reasonable to write a reward shaped specification of this form, but in most cases I think it would be much simpler to provide a few input/output examples, rather than to write a reward shaped spec.  Even in the case of sorting a list, providing input/output examples is pretty simple and natural and possibly easier for many less skilled users than coming up with the specification that you describe.

---

> > > > ### Author Response · Authors · 2018-01-04
> > > > **RE: Reponse**
> > > >
> > > > Thank you for your additional feedback. Regarding your remaining concerns:
> > > >
> > > > 1) We updated the paper and added results on program generalization to Table 3. Specifically, we reported success rates on 1000 held-out test cases for each task. We previously believed that manual inspection would be impractical because all programs in Table 3 are of length 100 and they are fairly complex. We decided that adding held out eval cases to each task is a better way to measure generalization, and we ran all of the synthesized code solutions on these additional test cases.
> > > >
> > > > 2) We agree with the reviewer that learning on input/output examples is simpler and more natural. We decided in this paper to write about single-task code synthesis because we wanted to focus on making that as good as possible, before moving on to the more complex multi-task scenario. It is not apparent to us how a single-task learning method can make use of test cases, since these test cases are static throughout training of the model.
> > > >
> > > > Though we use a shaped reward function, it is based on a Hamming distance function. We hope that it is generic enough to be usable on a wide array of coding tasks, and simple enough for less skilled users. Any set of input/output examples can be fed to this distance to create a proper reward function.

---

### Official Review · AnonReviewer3 · 2017-11-29
**The PQT algorithm is nice, but more analysis would be much appreciated. Also, BF is an odd language to target for program synthesis.**

**Rating:** 6
**Confidence:** 4

**Review:**

This paper introduces a method for regularizing the REINFORCE algorithm by keeping around a small set of known high-quality samples as part of the sample set when performing stochastic gradient estimation.

I question the value of program synthesis in a language which is not human-readable. Typically, source code as function representation is desirable because it is human-interpretable. Code written in brainfuck is not  readable by humans. In the related work, a paper by Nachum et al is criticized for providing a sequence of machine instructions, rather than code in a language. Since code in brainfuck is essentially a sequence of pointer arithmetic operations, and does not include any concept of compositionality or modularity of code (e.g. functions or variables), it is not clear what advantage this representation presents. Neither am I particularly convinced by the benchmark of a GA for generating BF code. None of these programs are particularly complex: most of the examples found in table 4 are quite short, over half of them 16 characters or fewer. 500 million evaluations is a lot. There are no program synthesis examples demonstrating types of functions which perform complex tasks involving e.g. recursion, such as sorting operations.

There is also an odd attitude in the writing of this paper, reflected in the excerpt from the first paragraph describing that traditional approaches to program synthesis “… typically do not make use of machine learning and therefore require domain specific knowledge about the programming languages and hand-crafted heuristics to speed up the underlying combinatorial search. To create more generic programming tools without much domain specific knowledge …”. Why is this a goal? What is learned by restricting models to be unaware of obviously available domain-specific knowledge?

All this said, the priority queue training presented here for reinforcement learning with sparse rewards is interesting, and appears to significantly improve the quality of results from a naive policy gradient approach. It would be nice to provide some sort of analysis of it, even an empirical one. For example, how frequently are the entries in the queue updated? Is this consistent over training time? How was the decision of K=10 reached? Is a separate queue per distributed training instance a choice made for implementation reasons, or because it provides helpful additional “regularization”? While the paper does demonstrate that PQT is helpful on this very particular task, it makes very little effort to investigate *why* it is helpful, or whether it will usefully generalize to other domains.

Some analysis, perhaps even on just a small toy problem, of e.g. the effect of the PQT on the variance of the gradient estimates produced by REINFORCE, would go a long way towards convincing a skeptical reader of the value of this approach. It would also help clarify under what situations one should or should not use this. Any insight into how one should best set the lambda hyperparameters would also be very appreciated.

---

> ### Author Response · Authors · 2017-12-22
> **Response**
>
> Thank you for your review. We address your key concerns:
>
> The reviewer is concerned with the value of generating code in BF, saying that it is not human-readable. We first want to point out that program synthesis is an important task by itself, without considering human readability. For example, having a way to reliably do program synthesis would help with algorithm induction problems (training a model to carry out an algorithm) where generalization past training domain is still an issue. Furthermore, we want the experimental setup we introduce in our paper to serve as MNIST for program synthesis. We believe that a method which can code in C++ or Python should at least be able to write code for BF. By starting with BF, we hope to remove many of the complexities of higher level languages while focusing on a core hard problem in the code synthesis space. We also want to note, that we do synthesize some programs which we consider to be human readable (see Table 4), by adding program length bonus to the reward. Though BF in general may be difficult to read, that does not mean code written in BF is useless.
>
> The reviewer is not convinced that domain-specific knowledge is a limitation. We agree that the reviewer is right in saying that we formulate our code synthesis problem in a very restrictive way that does not take advantage of task information or domain-specific knowledge. In an applied setting, like code completion, it would be in the experimenters' best interests to leverage all available knowledge and tools to solve the problem. In our case, however, our goal is to simplify the problem in order to isolate aspects of code synthesis that are fundamentally hard. We also want to show that general methods are viable for solving this problem, in the hope that they can be more easily adapted to any programming language, and might even benefit other applications of RL.
>
> The reviewer is concerned with the lack of analysis of why and how PQT works. We agree that it would be better to have included analysis about why and how PQT works. However, we wanted to keep the focus of the paper on the experimental setup and the comparison between methods. Understanding PQT and its effectiveness in other settings, as well as its benefits to REINFORCE, we leave to follow-up work.
>
> Regarding other concerns:
>
> The reviewer asks, "How was the decision of K=10 reached?" We would like to note that we say the following in the paper: "In early experiments we found 10 is a nice compromise between a very small queue which is too easy for the RNN to memorize, and a large queue which can dilute the training pool with bad programs." We also would like to add that we did not tune K, as the increase in hyperparameter search space would make tuning prohibitively expensive. So the choice of K here was more of an intuitive guess.
>
> The reviewer asks, "Is a separate queue per distributed training instance a choice made for implementation reasons, or because it provides helpful additional regularization?" We did indeed use a separate queue per distributed training instance to make the implementation easier. We update the paper to say that we use separate queues for ease of implementation.
>
> The reviewer comments, "Any insight into how one should best set the lambda hyperparameters would also be very appreciated." In the paper we discuss using grid search to tune hyperparameters, including lambda, and we give the search space we used. As with hyperparameters in many machine learning models, picking the correct values is a very difficult problem, and using standard hyperparameter tuning methods serves as a good first approximation.

---

### Author Response · Authors · 2017-12-22
**Revised manuscript to address reviewer feedback**

We thank all reviewers for their valuable feedback. To address the reviewers’ comments, we have revised the manuscript.

Here is a summary of the changes:

1) We update Table 3 to include eval results on a held-out test dataset (1000 test cases per task). This should give readers an idea of how well the synthesized programs generalize to each task.
2) We soften our claim on novelty of PQT and we give more credit to prior work, such as Liang et al 2017.
3) We also update Table 4 to highlight which code strings were observed to overfit the test cases via manual inspection.
4) In Section 3.2 we note that we use separate queues per worker for ease of implementation.
5) We fix our description of the test cases at the bottom of the section titled "Other model and environment choices." We also note differences between the tuning and eval variants of the "reverse" and "remove-char" tasks.

---

### Decision · Program_Chairs · 2018-01-29
**ICLR 2018 Conference Acceptance Decision**

**Decision:**

Reject

**Comment:**

This paper introduces a possibly useful new RL idea (though it's a incremental on Liang et al), but the evaluations don't say much about why it works (when it does), and we didn't find the target application convincing.